# Role of Biomarkers in the Prediction of Serious Adverse Events after Syncope in Prehospital Assessment: A Multi-Center Observational Study

**DOI:** 10.3390/jcm9030651

**Published:** 2020-02-28

**Authors:** Francisco Martín-Rodríguez, Carlos Del Pozo Vegas, Alicia Mohedano-Moriano, Begoña Polonio-López, Clara Maestre Miquel, Antonio Viñuela, Carlos Durantez Fernández, Jesús Gómez Correas, Raúl López-Izquierdo, José Luis Martín-Conty

**Affiliations:** 1Advanced Clinical Simulation Center, School of Medicine, Universidad de Valladolid. Advanced Life Support Unit, Emergency Medical Services, 47005 Valladolid, Spain; fmartin@saludcastillayleon.es; 2Emergency Department, Hospital Clínico Universitario de Valladolid, 47003 Valladolid, Spain; 3Faculty of Health Sciences. Universidad de Castilla la Mancha, 45600 Talavera de la Reina, Spain; Alicia.Mohedano@uclm.es (A.M.-M.); Clara.maestre@uclm.es (C.M.M.); Antonio.vinuela@uclm.es (A.V.); Carlos.durantez@uclm.es (C.D.F.); jesus.gomez@uclm.es (J.G.C.); JoseLuis.MartinConty@uclm.es (J.L.M.-C.); 4Emergency Department, Hospital Universitario Rio Hortega, 47012 Valladolid, Spain; rlopeziz@saludcastillayleon.es

**Keywords:** prehospital care, early warning scores, lactate, clinical decision, early mortality, clinical deterioration, syncope

## Abstract

Syncope is defined as the nontraumatic, transient loss of awareness of rapid onset, short duration and with complete spontaneous recovery, and accounts for 1%–3% of all visits to the emergency department. The objective of this study was to evaluate the predictive capacity of the National Early Warning Score 2 (NEWS2) and prehospital lactate (pLA), individually and combined, at the prehospital level to detect patients with syncope at risk of early mortality (within 48 h) in the hospital environment. A prospective, multicenter cohort study without intervention was carried out on syncope patients aged over 18 who were given advanced life support and taken to the hospital. Our study included a total of 361 cases. Early mortality affected 21 patients (5.8%). The combined score formed by the NEWS2 and the pLA (NEWS2-L) obtained an AUC of 0.948 (95% CI: 0.88–1) and an odds ratio of 86.25 (95% CI: 11.36–645.57), which is significantly higher than that obtained by the NEWS2 or pLA in isolation (*p* = 0.018). The NEWS2-L can help stratify the risk in patients with syncope treated in the prehospital setting, with only the standard measurement of physiological parameters and pLA.

## 1. Introduction

Syncope is defined as nontraumatic, transient loss of consciousness due to cerebral hypoperfusion, characterized by a rapid onset, short duration, and complete spontaneous recovery. Syncope itself represents a major challenge for both prehospital emergency medical services (EMS) and emergency department (ED) and accounts for 1%–3% of all ED visits [1,2,3].

Syncope is a symptom with many clinical presentations from different underlying causes. It may be the result of altered reflexes, causing recurrent episodes or manifestation of transcendent diseases leading to serious adverse events (SAEs) with a serious outcome, i.e., sudden cardiac death [4].

Different and multidisciplinary guidelines have been developed to manage syncope based on risk stratification [1,5,6].

However, in the prehospital context, clinical decisions should be made immediately, with only few complementary tests (vital signs, electrocardiogram, and capillary blood glucose), so EMS professionals should base their decisions on the most predominant symptom. In addition, the demography of the target population itself means that the majority of patients are older adults (>65 years) [7], with multiple pathologies and comorbidities, often after falls associated with syncope and in many cases with polypharmacy [8,9], i.e., highly complex patients.

The prehospital context is integrating different procedures and solutions that can help health professionals in the decision-making process, among them biomarkers, understood as the use of early warning scores (EWSs), point-of-care testing (POCT), or the combined use of both [10].

Within the EWSs, the National Early Warning Score 2 (NEWS2), is the most widely used internationally, is validated in the prehospital context, and has proven its usefulness in very diverse clinical contexts [11,12,13]. The NEWS2 is determined from simple clinical observations (respiration rate, oxygen saturation, supplemental oxygen, temperature, systolic blood pressure, heart rate, and level of consciousness). Aggregate weighting produces a final score and a level of risk that determines the emergency response (see Table 1). High scores are associated with an increase in prehospital advanced life support (PALS), admission to the intensive care unit (ICU), and early mortality [14].

One of the most promising biomarkers in the prehospital context, which can be employed at bedside, is the point-of-care lactate (pLA), a reliable indicator of hypoperfusion states caused by anaerobic metabolism [16,17,18].

There is a growing interest in analyzing the usefulness of NEWS2 and pLA, either in isolation [19,20], or together [21], but no studies have assessed the prognostic use of these biomarkers in patients with syncope, and less so in the prehospital context.

The objective of this study was to evaluate the predictive capacity of the NEWS2 and pLA (individually and in combination) at the prehospital level to detect patients with syncope at risk of early mortality (within 48 h) in the hospital environment. 

## 2. Material and Methods

### 2.1. Study Design

Between April 2018 and June 2019, we conducted a prospective, multicenter cohort study without intervention in adults over 18 years of age that were attended consecutively by the EMS and evacuated in advanced life support (ALS) to their reference hospitals with a main prehospital diagnosis of syncope.

The present study was approved by the Research Ethics Committee (REC) of the public health system of Castile and Leon (REC number: #PI 18-010, #PI 18-895, #PI 18-10/119, #PI MBCA/dgc, and #PI 2049). All patients (or guardians) signed informed consent. This study is reported in line with the STROBE statement [22].

### 2.2. Study Setting

This study was carried out in the EMS of the public health system of the Community of Castile and Leon (Spain), with the participation of six ALS distributed in the provinces of Burgos, Salamanca, Segovia, and Valladolid, with a reference population of 1,351,962 inhabitants. The EMS operate 24 h a day 365 days a year. Calls are received by a technician (nonhealth personnel), who collects administrative and geo-location data. A medical doctor (MD) filters the calls and determines the most appropriate resource for each situation. ALS are made up of two paramedics, an MD and an emergency registered nurse (ERN), performing standard advanced life support maneuvers at the scene or *en route.*

Three hundred, forty-three cases allow us to estimate the percentage of deaths expected by syncope, with an error not exceeding 2.5% (with a 95% confidence level). For this, we have assumed a loss percentage of around 15%.

All patients included in the study were referred to hospitals belonging to the public health system (Burgos University Hospital, Segovia Hospital Complex, Salamanca University Assistance Complex, Rio Hortega University Hospital, and Valladolid University Clinic), all of them with extensive surgical capacity and ICU.

### 2.3. Population

To identify eligible patients, the sample was recruited from among all calls for adult care demand (over 18 years old) during the study period, which required evacuation to the ED in ALS with a prehospital main diagnosis of syncope.

We excluded cases of cardiorespiratory arrest, patients in terminal status or with acute psychiatric pathology, pregnant women, patients evacuated by other means of transport or discharged in situ, patients in whom the first set of vital signs were incomplete and did not allow to calculate the NEWS2, and cases in which it has not been possible to determine pLA after two attempts. Before data analysis, we also excluded patients for whom it was not possible to obtain informed consent, and those who were assisted more than once (only the first chronological event was counted), or for whom followup via electronic medical record was not possible.

### 2.4. Study Protocol

The review protocol of this study was registered with ICTRP (doi.org/10.1186/ISRCTN17676798).

The principal investigator (FMR) trained all members of the research group on the objectives of the study, the standardized way of obtaining the set of vital signs, the use of electromedical equipment, and on the calculation and interpretation of the NEWS2 according to recommendations by The Royal College of Physicians [15]. Similarly, a procedure for determining pLA was developed, with specific training on the operation, cleaning, maintenance, and calibration of the equipment. The traceability of all test strips used in the study has been monitored, through the control of expiration date, serial number, and batch number.

A standardized case form was used (clinical history ordinarily used by the EMS), where the ALS ERN recorded the set of vital signs and pLA value. All prehospital clinical data analyzed refer to the first contact of the team with each patient. Next, the unit’s MD recorded demographic, times of arrival, assistance and evacuation, administrative, and other prehospital care variables: Electrocardiographic rhythm and need for PALS at the scene or during the transfer (orotracheal intubation, use of external pacemaker, or vasoactive drugs).

Within 30 days of the index event, an associated researcher from each hospital obtained hospital care data by reviewing the patient’s electronic medical history. This data includes hospital admissions and/or ICU and mortality within 48 h from the index event.

All patient data were recorded electronically in a database created for this purpose. Prior to statistical analysis, the database was cleaned by means of logical tests, range tests (for the detection of extreme values), and data consistency. Subsequently, the presence and distribution of unknown values of all variables were analyzed. The case registration form was tested to eliminate ambiguous elements and guarantee the robustness of the data collection instrument.

### 2.5. Data Abstraction

The main outcome variable was early hospital mortality (within the first 48 h) from any cause, in line with previous studies [11,12].

To calculate the NEWS2 (Table 1), we registered respiratory rate, temperature using a ThermoScan^®^ PRO 6000 tympanic thermometer (Welch Allyn, Inc, Skaneateles Falls, NY. USA), and oxygen saturation, systolic blood pressure and heart rate using a LifePAK^®^ 15 monitor (Physio-Control, Inc., Redmond, WA. USA). Mental state was assessed with the Glasgow Coma Scale (GCS). Confusion was defined as a GCS score of less than 15 points, or new confusion situation *en route*.

To obtain pLA values, we used an Accutrend Plus measuring device (Roche Diagnostics, Mannheim, Germany) with a measuring range of 0.8–21.7 mmol/l. The whole procedure consists of three phases: First, the test strip is inserted after switching on the instrument; second, a drop of venous blood (extracted in a 1-mL syringe) is deposited on the test strip (15–40 μL); and third, the lid is closed and a result is obtained after 60 s. Between blood collection and placement of the sample in the device, no more than 1 min should pass. All measuring devices were calibrated every 50 measurements, always by the same researcher, using the Accutrend^®^ BM-Control-Lactate control solution (Roche Diagnostics, Mannheim, Germany).

Secondary outcomes include advanced life support maneuvers during prehospital assistance and/or transfer and the need for ICU.

To obtain the combined value of NEWS2 and pLA, the numerical value of the pLA was added to the numerical value of the NEWS2, generating the new scale NEWS2-L.

### 2.6. Data Analysis

All data was stored in an XLSTAT^®^ BioMED database for Microsoft Excel^®^ version 14.4.0. (Microsoft Inc., Redmond, WA. USA), and SPSS version 20.0. (IBM, Armonk, NY. USA), which were also used for statistical analysis.

Quantitative variables are described as median and interquartile range (IQR) and qualitative variables are described by absolute frequencies with their 95% confidence interval (95% CI). The Mann–Whitney U-test was used to compare the locations of quantitative variables. The Chi-square test for 2 × 2 and/or contingency tables of proportions was used to determine the association or dependence relationship between qualitative variables; if necessary (percentage of boxes with expected values less than five greater than 20%), we used Fisher’s exact test.

The area under the curve (AUC) of the receiver operating characteristic (ROC) of the NEWS2, pLA, and for both in combination was calculated in terms of mortality at 48 h, PALS, and the need for admission to ICU. We also determined the score that offered greatest sensitivity and joint specificity in each case, as well as the positive predictive value (PPV), negative predictive value (NPV), positive probability ratio (PPR), and negative probability ratio (NPR) for these scores.

In all tests, a confidence level of 95% and a *p*-value of less than 0.05 were considered significant. The data are presented according to the Standards for Reporting Diagnostic Accuracy 2015 statement [23].

## 3. Results

### 3.1. Patient Baseline

Over the study period of 14 months, 720 cases were screened for eligibility and 361 patients fulfilled all inclusion criteria (Figure 1). 

Median age was 74 years (IQR 62–83 years), 164 (54.4%) of the participants were women. The median times of arrival, assistance, and evacuation were 9 min (IQR 7–11 min), 28 min (IQR 22–35 min), and 9 min (IQR 6–12 min), respectively, without significant differences between survivors and nonsurvivors. Characteristics of the patients who were diagnosed with syncope in the prehospital setting are presented in Table 2.

After prehospital care, 5.8% (21 cases) of patients with syncope died in less than 48 h. In 61.9% (13 cases) the main cause of mortality was cardiovascular pathologies, followed by infectious and neurological processes. The need for PALS affected 14.1% (51 cases), 16 (76.2%) in nonsurvivors, and 35 (10.3%) in survivors (*p* < 0.001). ICU was required by 52.4% (11 cases) of nonsurvivors (*p* < 0.001) (Table 2).

Nonsurvivors were significantly older, with high scores of NEWS2 and pLA, and presented more PALS and admission to ICU.

### 3.2. Prognostic Accuracy of the Scores

The prognostic precision of the scores for predicting early mortality is represented in Figure 2a. The NEWS2-L, with an AUC of 0.948 (95% CI: 0.88–1) had the best performance score. A comparison of the curves was statistically significant for the NEWS2-L with respect to the other scores studied (*p* = 0.018).

With respect to the early detection in PALS, the best performing score was also the NEWS2-L, with an AUC of 0.842 (95% CI: 0.77–0.91), although it did not reveal significant differences with the NEWS2 (comparison of curves *p* = 0.540). Figure 2b displays the different AUC with their confidence intervals and *p*-value.

For predicting the risk of the need for ICU, the NEWS2, pLA, and NEWS2-L provided comparable results (*p*-value for all scales greater than 0.05). The score with the best prognostic capacity was again the NEWS2-L with an AUC of 0.809 (95% CI: 0.71–0.90), as can be seen in Figure 2c.

### 3.3. Cut-off Points of the NEWS2-L

The endpoints of 48-h mortality, PALS, and ICU obtained 9.5 points, 6.9 points, and 10.3 points, respectively. The capacity of the NEWS2-L to predict 48-h mortality with an odds ratio of 86.25 (95% CI: 11.36–645.57) and a NLR of 0.06 (95% CI: 0.01–0.40) stand out. For all outcomes studied, very high NPV were maintained. The results for the different outcomes for the NEWS2-L can be seen in Table 3.

## 4. Discussion

In this multicenter observational cohort study, we found that the NEWS2-L presented a better behavior, equally for cases of early mortality, PALS, and the need for ICU.

### 4.1. Comparison with Previous Studies

Our results are in line with studies that have analyzed prognostic scores and mortality data for syncope patients [24,25,26,27,28], although those had been conducted in the ED. Other work has analyzed the combined use of EWSs and lactate in other contexts [29,30,31,32] and also obtained areas under the ROC curve between 0.830 and 0.914 for detecting two-day mortality. Our data confirm and add to previous studies, being the only study specifically developed to detect patients with syncope at high risk in the prehospital context, which provides very useful information on the need for prehospital advanced life support or ICU. 

### 4.2. Managing Prehospital Syncope

Current guidelines for the management of syncope emphasize the initial clinical and physical evaluation, as well as the early performance of an electrocardiogram and analytical tests, with the fundamental objective of not underestimating potentially risky syncope situations [1,4,33,34]. The new score, NEWS2-L, is in line with the decision rules implemented in the current guidelines, with a high specificity (81.2%; 95% CI 76.7–85.0), higher than other risk stratification scores, such as the FAINT score [35] (22.2%; 95% CI 20.7–23.8), HEART score [26] (36.5%; 95% CI 33.0–40.2), or the OESIL score [36] (73.0%; 95% CI 0.66–0.74). This makes this score an ideal tool for the detection of syncopal episodes that could initially be classified as mild but that actually present a high risk of deterioration. Therefore, we consider that introducing both the NEWS2 and the NEWS2-L in the management recommended by the current guidelines could aid decision-making in the first contact with the patient, both when assessing the true severity of the condition as when identifying the need for a hospital transfer [37,38].

### 4.3. Implications

Due to the ease of application and interpretation of the NEWS2-L, this score is ideal for use by nonexpert health personnel and can help discriminate the severity of the condition in different contexts very early. Therefore, it could be an advantage to not include electrocardiogram interpretation, more specific analytical tests or the presence of guiding symptoms, as in the FAINT score (history of heart failure, history of cardiac arrhythmia, initial abnormal ECG result, elevated pro B-type natriuretic peptide, and elevated high-sensitivity troponin T) [35], HEART (history, ECG, age, risk factors, and troponin) [26], OESIL score (abnormal ECG, age >65 years, history of cardiovascular disease) [39].

The social alarm that is generated when a person suffers a syncope, often due to the florid and unspecific symptoms that precede or accompany this condition (decreased level of consciousness, chest pain, seizures or muscle stiffness, vomiting, profuse sweating, pale skin, incontinence, etc.), makes this pathology one of the most frequent causes to which EMS must respond [40,41].

Our data indicate that a NEWS2-L score of 9.5 or higher implies with high probability a risk of death within 48 h, and a score of 6.9 or higher was associated with a higher frequency of advanced life support maneuvers at the scene or on the road, which means it can help us detect high-risk patients who may need advanced maneuvers. 

### 4.4. Limitations

This study has several potential limitations. The first limitation is the possible patient selection bias, as the study was carried out based on opportunity criteria for a limited period of time and only in those patients with a prehospital diagnosis of syncope who were evaluated and evacuated in ALS.

Second, the main outcome variable is hospital mortality from any cause within the first 48 h. Deaths occurring outside this temporary window or outside the hospital have not been counted. For future studies, it may be very interesting to analyze medium and long-term mortality, as well as extra-hospital mortality, but for this study we have considered this time limit for being the deaths that occurred acutely.

Finally, the small sample size allows for preliminary results, but is insufficient to perform an external validation of the score. Hence, it would be necessary to conduct prospective multicenter studies with adequate power in different geographical contexts and with different EMS to validate the routine use of the NEWS2-L in the prehospital context for the early detection of impairment in patients with syncope.

## 5. Conclusions

In summary, the NEWS2-L can help identify patients who have undergone syncope and have a high-risk of needing ICU or early mortality in the prehospital setting, with only the standard measurement of physiological parameters and pLA, but it is a necessary more elaborate diagnostic evaluation to stratify the risk of patients after syncope to allow personalized management of different treatment options.

EMS must implement among its routine procedures for the use of EWS and point of care tests to assist in decision making in clinical processes.

## Figures and Tables

**Figure 1 jcm-09-00651-f001:**
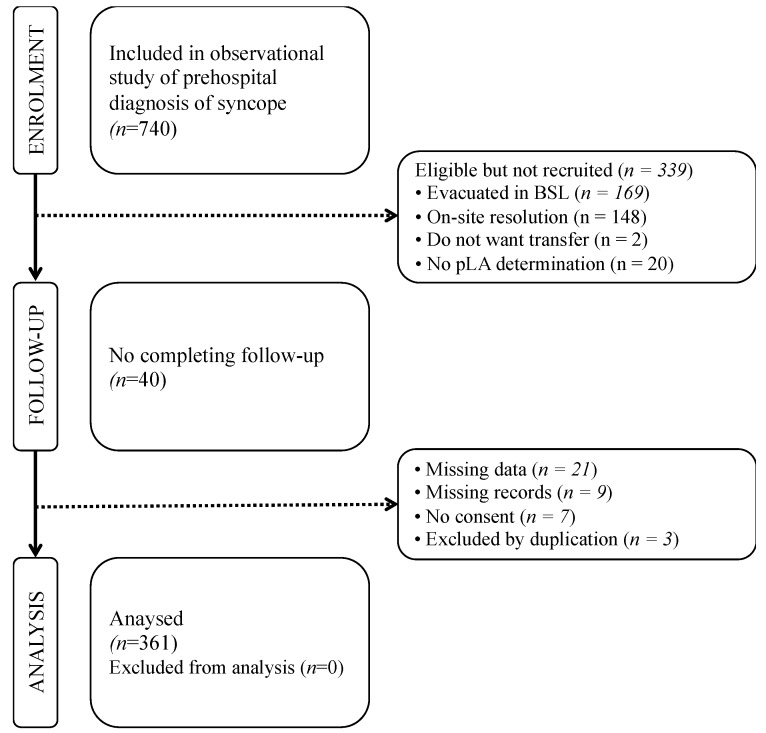
Flowchart of the participants in the study. BLS: Basic life support.

**Figure 2 jcm-09-00651-f002:**
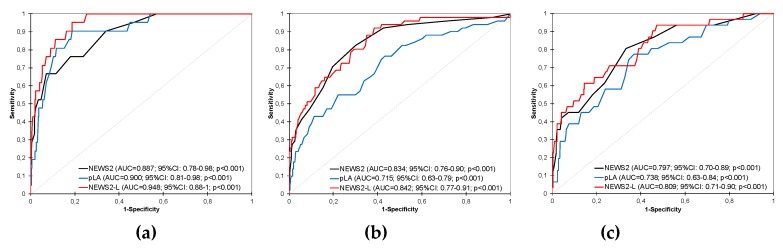
Diagnostic performance curves and areas under the curve with 95% confidence intervals for NEWS2, pLA, and NEWS2-L. **(a)** For two-day mortality; **(b)** for prehospital advanced life support (PALS); **(c)** for intensive care unit (ICU). NEWS2: National Early Warning Score 2; pLA: Point-of-care lactate; NEWS2-L: National Early Warning Score 2 with point-of-care lactate.

**Table 1 jcm-09-00651-t001:** National Early Warning Score 2 (NEWS2).

NEWS2	3	2	1	0	1	2	3
Pulse (bpm)	≤40		41–50	51–90	91–110	111–130	≥131
BR (bpm)	≤8		9–11	12–20		21–24	≥25
T (°C)	≤35		35.1–36	36.1–38	38.1–39	≥39.1	
SBP (mmHg)	≤90	91–100	101–110	111–219			≥220
SpO2 (%) Scale 1	≤91	92–93	94–95	≥96			
SpO2 (%) Scale 2 ^1^	≤83	84–85	86–87	88–92 ≥93 air	93–94 Oxygen	95–96 Oxygen	≥97 Oxygen
Air oxygen		Oxygen		Air			
AVPU (scale)				A			V, P, U

^1^ In patients with hypercapnic respiratory insufficiency, scale 2 should be used to weight the oxygen saturation score. Each category is graded 0–3. Scores for each category are added to give a total. Composite scores of greater than 5 (or 3 in any one parameter) trigger an urgent medical review. A score of over 7 triggers a review by a critical care outreach team or medical response team [15]. SBP: Systolic blood pressure; BR: Breathing rate; SpO2: Oxygen saturation; AVPU: Alert, verbal, pain, unresponsive; T: temperature; GCS: Glasgow coma scale. Reproduced from: Royal College of Physicians. National Early Warning Score (NEWS) 2: Standardising the assessment of acute-illness severity in the NHS. Updated report of a working party. London: RCP, 2017 [15]

**Table 2 jcm-09-00651-t002:** Demographic, prehospital, and hospital data of included patients (death statistics refer to mortality rates in less than 48 h).

Characteristic ^1^	Total	Survivors	Non-Survivors	*p*-Value
Number [*n* (%)]	361 (100)	340 (94.2)	21 (5.8)	
Age (years) [*Median* (IQR)]	74 (62–83)	74 (62–83)	81 (73–87)	0.010 ^2^
Female [*n* (%)]	164 (45.4)	155 (94.5)	9 (5.5)	0.807 ^4^
Initial prehospital evaluation values (*Median* (IQR))				
NEWS2 (points)	3 (1–6)	3 (1–5)	11 (6–15)	0.001 ^2^
Breathing rate (bpm)	16 (12–19)	16 (12–19)	20 (16–32)	0.001 ^2^
Oxygen saturations (%)	96 (94–98)	96 (94–98)	89 (76–94)	0.001 ^2^
Temperature (°C)	36.0 (35.8–36.6)	36.1 (35.8–36.6)	35.7 (34.9–36.2)	0.012 ^2^
SBP (mmHg)	127 (106–144)	128 (108–144)	105 (84–139)	0.132 ^2^
Heart rate (bpm)	71 (56–87)	70 (57–87)	82 (40–96)	0.308 ^2^
GCS (points)	15 (15–15)	15 (15–15)	14 (7–15)	0.006 ^2^
*p*LA (mmol/l)	2.7 (1.9–3.8)	2.6 (1.8–3.6)	5.2 (4.5–6.6)	<0.001 ^2^
Blood glucose (mg/dl)	128 (109–159)	127 (107–155)	177 (118–260)	0.006 ^2^
NEWS2-L	6.2 (3.9–9.1)	5.9 (3.8–8.8)	16.4 (12.2–21.0)	<0.001 ^2^
Electrocardiographic rhythm [*n* (%)]				
Sinus	197 (54.6)	192 (56.5)	5 (23.8)	
Sinus bradycardia	47 (13.0)	46 (13.5)	1 (4.8)	
Sinus tachycardia	40 (11.1)	33 (9.7)	7 (33.3)	
Atrial fibrillation	35 (9.7)	34 (10.0)	1 (4.8)	
Atrioventricular block	30 (8.3)	24 (7.1)	6 (28.6)	
Pacemaker pace	12 (3.3)	11 (3.2)	1 (4.8)	0.097 ^3^
Prehospital advanced life support [*n* (%)]				
Orotracheal Intubation	11 (3.0)	4 (1.2)	7 (33.3)	0.001 ^4^
Vasocative drugs	38 (10.5)	25 (7.4)	13 (61.9)	0.001 ^4^
External pacemaker	24 (6.6)	18 (5.3)	6 (28.6)	0.001 ^4^
Hospital outcomes [*n* (%)]				
Inpatients	137 (38.0)	116 (34.7)	21 (100)	0.001 ^4^
Intensive care unit	31 (8.6)	20 (5.9)	11 (52.4)	0.001 ^4^

^1^ Values expressed as total number (fraction) and medians [25th percentile–75th percentile] as appropriate. ^2^ The *p*-values were calculated with Mann–Whitney U-test. ^3^ For the calculation of the *p*-value with Chi-square test, the rhythms have been regrouped into two sets, one sinus rhythm, sinus tachycardia and bradycardia, and another set with atrial fibrillation, atrioventricular block, and pacemaker pace. ^4^ The *p*-values were calculated with Chi-square test. IQR: Interquartile range; SBP: Systolic blood pressure; GCS: Glasgow coma scale; pLA: Point-of-care lactate.

**Table 3 jcm-09-00651-t003:** Cut-off points for combined sensitivity and specificity with best score (Youden’s test) on the NEWS2-L for two-days mortality, PALS, and ICU.

	Two-Days Mortality ^1^	PALS^ 1^	ICU ^1^
Prevalence	0.058	0.141	0.086
Cut-off	9.5	6.9	10.3
Se	95.2 (77.3–99.2)	92.2 (81.5–96.9)	61.3 (43.8–76.3)
Sp %	81.2 (76.7–85.0)	61.6 (56.1–66.9)	85.5 (81.2–88.9)
PPV	23.8 (16.0–33.9)	28.3 (22.0–35.6)	28.4 (19.0–40.1)
NPV	99.6 (98.0–99.99)	97.9 (94.8–99.2)	95.9 (93.0–97.6)
LR(+)	5.06 (3.98–6.44)	2.40 (2.04–2.82)	4.21 (2.87–6.18)
LR(-)	0.06 (0.01–0.40)	0.13 (0.05–0.33)	0.45 (0.29–0.71)
OR	86.25 (11.36–645.57)	18.26 (6.62–53.69)	9.30 (4.24–20.39)
DA	82.0 (77.7–85.6)	65.9 (60.9–70.6)	83.4 (79.2–86.9)

^1^ Bracketed numbers indicate 95% confidence interval. PALS: Prehospital advanced life support; ICU: Intensive care unit; Se: Sensitivity; Sp: Specificity; PPV: Positive predictive value; NPV: Negative predictive value; LR: Likelihood ratio; OR: Odds ratio; DA: Diagnostic accuracy.

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
