# Peer review of "Role of Biomarkers in the Prediction of Serious Adverse Events after Syncope in Prehospital Assessment: A Multi-Center Observational Study"

_jcm, 2020, doi:10.3390/jcm9030651_

Round 1

Reviewer 1 Report

The authors propose a tool for prediction of serious adverse events after syncope in prehospital assessment. The main idea of the paper is very clear and the investigation of recognizing patients with low and high risk condition is important. The authors evaluate the predictive capacity of two markers individually and in combination. They used correctly statistical methods, however basic and simple. The scientific work presented in the paper is interesting and the results are valuable. Nonetheless, major revision is needed, and the following points should be considered:

1)Mann-Whitney test is not appropriate for comparing means. Why you did not use the parametric tests?

2)The “percentage of boxes with expected values less than 5 greater than 20%” is a recommendation for using Fisher’s exact test?

3)The descriptive statistics of the data should also include the mean and sd or 95%CI

4)When you present the results, you should depict at the beginning the name of statistical test that gives You the p-value.

5)Explain and discuss the 95CI: 11.36-645.57 for NEWS-L to predict 48-hour mortality

6) Did You perform sample size calculation? Please describe shortly the results

7) The data were collected properly but You should comment and discuss the fact, that in statistical analysis there were performed simple comparisons between two groups with different size (340 vs 21).

8)The statistical methods used for the analysis are very simple. Such large group of patients may vary according to many anthropometric parameters and other conditions that can affect the mortality risk. It would be interesting to build a model indicating the significant parameters that can also calculate the mortality risk. The sample is large enough to perform such analysis.

Author Response

Point 1: Mann-Whitney test is not appropriate for comparing means. Why you did not use the parametric tests?

Response 1: We agree with the reviewer that the Mann-Whitney test does not compare equality of means, so we have modified the sentence in the statistical methods section by: “The Mann-Whitney U-test was used to compare the locations of quantitative variables "

Preventively, thinking that one of the groups is very small (21 cases) we have systematically used the Mann-Whitney U-test because it avoids us having to assume the normality of the quantitative variables, since the group of 21 cases is very Small and even normality tests may not detect significant deviations from normality.

Point 2: The “percentage of boxes with expected values less than 5 greater than 20%” is a recommendation for using Fisher’s exact test?

Response 2: In table 2, to avoid the artifact mentioned by the Chi-square reviewer, it was applied to a modified version of the table, corresponding to merging similar rows according to a clinical criterion, namely sinus rhythm has been combined with sinus tachycardia and bradycardia . On the other hand, a unique group has been made with the rest of the electrocardiographic rhythms that generate arrimias (atrial fibrillation, pacemaker pace and atrio-ventricular blockages). After this regrouping, the Chi-square we offer in the table has been recalculated (0.097). We have left the complete table so that the reader can observe the distribution of the rhythms. An explanatory note has been introduced at the bottom of table 2.

Point 3: The descriptive statistics of the data should also include the mean and sd or 95%CI

Response 3: We have decided to summarize the location with medians and variability with IQR, both can be considered as standard measures for the objectives of the study and we consider that it is sufficient for the reader to have a clear idea of the description of the sample.

Point 4: When you present the results, you should depict at the beginning the name of statistical test that gives You the p-value.

Response 4: We have introduced relevant clarifications in the tablet 2 in the table footer.

Point 5: Explain and discuss the 95CI: 11.36-645.57 for NEWS-L to predict 48-hour mortality

Response 5: Although it seems a very large interval, the interval is useful, because it tells us that the risk is at least 10 times larger in the individuals who have tested positive, even though we cannot quantify exactly in that range what the true value is.

Point 6: Did You perform sample size calculation? Please describe shortly the results

Response 6: We have added the following sentence in the Study Setting and Population section to explain how the sample size was calculatedThree hundred, forty-three cases allow us to estimate the percentage of deaths expected by syncope, with an error not exceeding 2.5% (with a 95% confidence level). For this, we have assumed a loss percentage of around 15%.

Point 7: The data were collected properly but You should comment and discuss the fact, that in statistical analysis there were performed simple comparisons between two groups with different size (340 vs 21).

Response 7: It is true, but the statistical methods used allow this type of comparison with unbalanced groups. It is a descriptive and prospective work in which the data have been collected consecutively and the two cohorts of patients, survivors versus non-survivors have been analyzed and one of the limitations of this work is the low number of patients in the group of non-survivors for the comparison of the variables, but with the sample size obtained, we believe that it is representative enough.

Point 8: The statistical methods used for the analysis are very simple. Such large group of patients may vary according to many anthropometric parameters and other conditions that can affect the mortality risk. It would be interesting to build a model indicating the significant parameters that can also calculate the mortality risk. The sample is large enough to perform such analysis.

Response 8: Statistical methods are common in this type of studies in which it is intended to analyze the usefulness of a diagnostic test such as the NEWS scale and the NEWS-L scale. We appreciate your suggestion for the creation of a predictive model, but that was not the primary objective of the work.

Reviewer 2 Report

The authors conducted a multicenter study to evaluate whether the NEWS2 score and prehospital lactate levels as well as the combination of the two (NEWS2-L) are useful scores to predict high risk (early death, need for advanced life support and ICU admission) in patients with syncope at a prehospital level.

They found that the NEWS2-L score presents a better prognostic precision than the original NEWS2 score or the prehospital lactate level.

The study seems well conducted, and the conclusions are appropriate. However, the follow-up time of 48h for a serious outcome is  quite short, which is also stated by the authors in the limitations section.

I have a few additional questions / comments:

  • Did the authors perform some kind of sample size calculation? If not, how was the duration of the study determined.
  • How many patients were excluded due to missing lactate levels? Did sample collection delay assistance time?
  • How did assistance time compare to other studies?
  • Please add NEWS2-L score to Table 2

Author Response

Response to Reviewer 2 Comments

Point 1: The study seems well conducted, and the conclusions are appropriate. However, the follow-up time of 48h for a serious outcome is  quite short, which is also stated by the authors in the limitations section.

Response 1: We agree with the reviewer that it may be a short time, but a fundamental objective of the study is to detect situations of risk of deterioration early, and we have considered that two-days mortality is an indicated measure and is in line with similar studies. However, in the limitations section we have included the following sentence: "but for this study we have considered this time limit for being the deaths that occurred acutely "

Point 2: Did the authors perform some kind of sample size calculation? If not, how was the duration of the study determined.

Response 2: We have added the following sentence in the Study Setting and Population section to explain how the sample size was calculatedThree hundred, forty-three cases allow us to estimate the percentage of deaths expected by syncope, with an error not exceeding 2.5% (with a 95% confidence level). For this, we have assumed a loss percentage of around 15%”.

Point 3: How many patients were excluded due to missing lactate levels? Did sample collection delay assistance time?

Response 3: Twenty cases were excluded from the study because it was not possible to determine pLA values after two attempts. This information has been added in the flowchart and Figure 1 has been modified.

In no case the health care required by the sick person was delayed by performing the test, the placement of the venous line was used, and with the rest of the blood that is deposited in the posterior chamber of the angiocateter was the test performed.

Point 4: How did assistance time compare to other studies?

Response 4: Attendance time is a very important variable that we have assessed, but unfortunately it is not possible to compare it with other similar studies for several reasons: the EMS in Spain is made up of medical doctor, nurse and two paramedics, performing very advanced stage actions , differential fact with other EMS, and that makes the assistance systems in this special pathology hardly comparable. Our maxim is "to be and do" and not "to load and run", obviously if the patient's pathology requires it, and always without unreasonably delaying the arrival to the emergency department (eg in a patient with suspected hypovolemic shock the priority is the urgent transfer). For all these reasons, the comparison of assistance times, although it seems to be a factor to consider, has not been possible to compare it with other systems.

Point 5: Please add NEWS2-L score to Table 2

Response 5: The statistics requested by the reviewer have been calculated and included in table 2.

Round 2

Reviewer 1 Report

The authors responded to my comments satisfactorily. Nonetheless, minor revision is needed, and the following points should be considered:

Ad.point.3

The descriptive statistics of the data should also include the mean and sd or 95%CI. Please add the table with the mean and sd or 95% CI for continuous data.

Ad.point.8

The predictive models are commonly used in such type of analysis and they can bring a new insight on the analyzed problem. Please add in the limitations, that You did not analyze the parameters that could affect the mortality risk.

Author Response

Ad.point.3

The descriptive statistics of the data should also include the mean and sd or 95%CI. Please add the table with the mean and sd or 95% CI for continuous data.

Response Ad.point.3: According to the reviewer's recommendations, the mean and standard deviation for all continuous variables have been added in table two.

Additionally, the appropriate explanations and the meaning of the new abbreviations included have been added at the bottom of the table.

Ad.point.8

The predictive models are commonly used in such type of analysis and they can bring a new insight on the analyzed problem. Please add in the limitations, that You did not analyze the parameters that could affect the mortality risk.

Response Ad.point.8: We agree with the reviewer and understand your reservations. The methodology of predicted model analysis is very powerful and useful, but we have not used it in this work because the NEWS2 scale itself is already a predictive model. However, for future work we will explore this analysis technique.

In the limitations section we have added the following sentence, to influence the reviewer's idea and we hope that now it becomes clearer:

"Finally, the use of predictive models is a common practice in this type of analysis, but it was decided not to use this type of methodology as the NEWS2 scale is a model in itself. Similarly, it should be noted that no parameters were analyzed that may affect the risk of mortality, such as comorbidities (Charlson comorbidity scale), for future studies we intend to evaluate these factors".
